# Comparative Performance of Citrate, Borohydride, Hydroxylamine and β-Cyclodextrin Silver Sols for Detecting Ibuprofen and Caffeine Pollutants by Means of Surface-Enhanced Raman Spectroscopy

**DOI:** 10.3390/nano10122339

**Published:** 2020-11-25

**Authors:** Michele Lemos de Souza, Juan Carlos Otero, Isabel López-Tocón

**Affiliations:** 1Instituto de Ciências Exatas, Universidade Federal Fluminense, Volta Redonda 27213-145, Rio de Janeiro, Brazil; 2Andalucía Tech, Unidad Asociada IEM-CSIC, Departamento de Química Física, Facultad de Ciencias, Universidad de Málaga, E-29071 Málaga, Spain; jc_otero@uma.es

**Keywords:** SERS, β-cyclodextrin, Ag nanoparticles, ibuprofen, caffeine, trans-cinnamic acid

## Abstract

The detection of emerging contaminants in the aquatic environment, such as ibuprofen and caffeine, was studied by means of surface-enhanced Raman spectroscopy (SERS) using Ag nanoparticles (AgNPs) synthesized with β-cyclodextrin (βCD) as a reducing agent. The effect on the SERS signal of different molar ratios of Ag^+^/βCD in the synthesis route and the aging process of AgNPs were investigated by using trans-cinnamic as a test molecule. The SERS effectiveness of these β-cyclodextrin colloids (Ag@βCD) was also checked and compared with that of other silver sols usually employed in SERS synthesized by using other reducing agents such as citrate, borohydride and hydroxylamine. All the synthesized SERS substrates were characterized by different techniques. The experimental results indicate that Ag@βCD with the more diluted Ag^+^/βCD molar ratio showed the best SERS signal, enabling detection at trace concentrations of 0.5 µM in the case of trans-cinnamic acid. The Ag@βCD sols also showed the best sensitivity for detecting ibuprofen and caffeine, reaching the lowest limit of detection (0.1 mM). The proposed synthetic route for Ag@βCD sols provides an improved SERS substrate for detecting organic pollutants with better performance than other standard silver sols.

## 1. Introduction

Different chemical and physical methods are employed in processing freshwater from rivers or natural reservoirs and in water treatment plants in order to provide clear drinking water to the final consumer. However, due to irregular untreated sewage disposal, many organic contaminants of anthropogenic origin (the so-called emerging contaminants, ECs) have been frequently detected in the aquatic environment, which cannot be properly removed in the treatment plants. The ECs represent newly identified chemical contaminants that have not been covered by regulatory programs or routine monitoring, becoming an attractive issue for the scientific community [1,2].

One can find a great variety of sources of ECs such as drugs, hormones, perfluorinated and perchlorate compounds, flame retardants, industrial additives, personal care products, etc., which are constantly introduced into the aquatic environment, therefore being considered pseudopersistent [3,4,5]. Some ECs present potential threats not only to the aquatic environment but also to human health, given that they can act as endocrine disruptors, i.e., exogenous substances that act in the endocrine system (hormonal system), causing changes in physiological functions that impair the functioning of healthy organisms through adverse hormonal effects.

The current analysis methods for ECs are commonly based on different separation treatments by gas chromatography or by high-performance liquid chromatography associated with spectroscopic analysis such as UV-vis, fluorescence and mass spectroscopy. However, these techniques usually require rigorous sample pre-treatment process and the use of a large volume of organic solvents, becoming time-consuming processes. Hence, the development and improvement of detection methods for ECs are highly desirable at trace level concentrations, whereas the main difficulties in the analysis lie in the complexity of the environmental matrices and their emerging nature [6,7,8].

Surface-enhanced Raman scattering (SERS) [9,10,11] is a useful spectroscopic technique for molecular recognition through the characteristic vibrational bands of pollutants and aromatic molecules [12,13] at trace level concentrations, due to the enormous enhancement of Raman signals when a molecule is close to a rough metal surface of nanometric size. This technique has the advantages of high molecular selectivity, and no pre-treatment process is required [14,15].

A great variety of SERS substrates can be employed such as metallic powders or films, electrochemically activated metallic surfaces or, the most frequently used, colloidal solutions. A major problem is that the SERS signal is very dependent on the nature of the substrate as well as of the particular molecule, and, consequently, many SERS studies are focused on modifying the synthetic route for these colloids with the aim of improving their sensitivity. Another strategy is to functionalize the nanoparticles (NPs) for detecting aromatic molecules with poor solubility in water, for instance [16,17]. Different modified sensors have been proposed [18,19,20], most of them based on the use of specific molecules such as calixarenes [21], viologen dications [16,17] or cyclodextrins [22,23,24,25] that can act as molecular hosts of specific analytes. All these molecules show an appropriate structure, where host–guest interaction is possible, allowing the analyte to be close to the NP’s surface.

The aim of this work was to improve the SERS detection of ECs by modifying the original synthetic route for silver colloids based on the use of β-cyclodextrin (βCD) as a reducing agent [26]. βCD is a cyclic oligosaccharide with seven D-glucose units linked by α(1–4) bonds, resulting in a toroid cone structure of about 7 nm in diameter, with a hydrophobic core and a hydrophilic exterior [27]. Moreover, it can be used as a reducing agent in alkaline medium and also allows for stabilizing the Ag@βCD colloid in the bulk as was firstly reported by Pande et al. [26]. Changes in the Ag^+^/βCD molar ratio, the addition of stabilizers or stronger reducing agents have been proposed in order to obtain a better SERS signal [22,23,24,25]. Therefore, βCD can play a triple role: as a reducing agent for Ag^+^ cations, as a stabilizer of the colloid and as a modifier of the metallic surface, which can be partially functionalized with CDs allowing for host–guest interactions for particular adsorbates.

In this work, the SERS effectiveness of the three types of Ag nanoparticles (AgNPs) synthesized with βCD were checked and compared with those usually employed in SERS by using other reducing agents such as citrate, borohydride and hydroxylamine. For this purpose, trans-cinnamic acid was selected as a probe molecule, given that it provides strong SERS spectra [28]. These AgNPs were employed in the study of the detection of the persistent organic pollutants ibuprofen (IBU), a non-steroidal anti-inflammatory drug, and caffeine (CAF), a nitrogenous organic compound of the alkaloid group. Both of them are indicators of polluted water sources. Appendix A shows the pictorial representation and the B3LYP/6-31G* optimized structure of the studied molecules. The limit of detection was determined as the lowest concentration of the sample whose characteristic vibrational bands could be recognized in the SERS spectra.

## 2. Materials and Methods

### 2.1. Materials

The reagents (ibuprofen sodium salt (CAS 31121-93-4,), caffeine (CAS 58-08-2), trans-cinnamic acid (CAS 140-10-3), β-cyclodextrin (CAS 7585-39-9), trisodium citrate dihydrate (C_6_H_5_Na_3_O_7_.2H_2_O, CAS 6132-04-3), sodium borohydride (NaBH_4_, CAS 16940-66-2) and hydroxylamine hydrochloride (NH_2_OH.HCl, CAS 5470-11-1)) were analytical grade and used without further purification. Sodium sulfate, Na_2_SO_4_, and silver nitrate, AgNO_3_, were also of the highest purity available. All the products were purchased from Aldrich (Sigma-Aldrich Inc., San Luis, MO, USA). The stock solutions and silver sols were prepared with water from a Milli-Q system (18.2 MΩ.cm resistivity at 25 °C). All the glassware was cleaned with a HCl (37%)/HNO_3_ (60%) solution in a 3:1 ratio.

### 2.2. Synthesis Procedure for AgNPs

#### 2.2.1. AgNPs Using Standard Reducing Agents

The synthetic procedure for obtaining AgNPs by reducing a silver nitrate solution with different agents such as citrate (Ag@Cit) [29], borohydride (Ag@BH) [29,30,31] and hydroxylamine hydrochloride (Ag@HX) [12,32,33] has already been reported elsewhere.

Briefly, the preparation of the Ag@Cit NPs was performed by heating 100 mL of a 1 mM AgNO_3_ aqueous solution until it started to boil, and then, 2 mL of a 1% trisodium citrate dihydrate solution was added. It was kept under heating and magnetic stirring for 1 h, approximately. The final colloidal solution was characterized by a dark-yellow/greenish-gray-colored solution.

The Ag@HX NPs were prepared at room temperature under vigorous magnetic stirring by adding, drop by drop, 10 mL of 10 mM AgNO_3_ solution into 90 mL of 16 mM hydroxylamine aqueous solution in an alkaline medium (300 µL of NaOH, 1 M). It was kept under stirring for 15 min once the addition of silver nitrate was complete. A grayish-green-colored colloidal suspension was obtained.

The Ag@BH NPs were prepared by adding, dropwise, 10 mL of 1 mM AgNO_3_ aqueous solution into 40 mL of 3 mM borohydride aqueous solution under vigorous magnetic stirring in an ice bath. Manual stirring was employed at the end of the process, and the clear, light-yellow colloidal suspension turned into a caramel-colored sol.

All the colloidal solutions were kept for a day and also five days before using them for the SERS measurements in order to evaluate the presence of residual reductants and the aging effect.

#### 2.2.2. AgNPs Synthesized with βCD

Taking the original experiment by Pande et al. [26] as a reference, two new syntheses were planned in this work. Both had the same molar ratio of Ag^+^/βCD, but one lower than that published [26], and one of them was ten times more diluted than the other one. Thus, three syntheses were performed, denoted as Ag@βCD1, Ag@βCD2 and Ag@βCD3.

Ag@βCD1 was prepared following the published procedure [26]. Briefly, 0.396 g of βCD was dissolved in 49.3 mL of water, and the solution was kept under stirring for 10 min. Then, 200 µL of 10 mM AgNO_3_ solution was added, and the mixture was kept under stirring for an additional 2 min. After that, 0.5 mL of 1 M NaOH solution was added, making an alkaline medium with pH = 10–12. The solution was then heated in a water bath (90 °C) for 20 min. The Ag^+^/βCD molar ratio of this colloid was 1:173.

The Ag@βCD2 synthesis was modified with respect to the above procedure. Briefly, 30 mL of 15 mM βCD solution was diluted by adding 15 mL of water and 15 mL of 10 mM NaOH. The solution was kept under heat in a water bath (60 °C) and magnetic stirring for 5 min. Then, 20 mL of 10 mM AgNO_3_ solution was added to the heated solution drop by drop, slowly. The final solution was kept under heating and vigorous magnetic stirring for an additional 10 min. The final solution was dark grayish green, similar to Ag@HX. The Ag^+^/βCD molar ratio of this colloid was 1:2.25.

Ag@βCD3 was made following the above procedure, but the concentrations of the employed precursors were ten times more diluted. In this case, the concentrations were 1.5 mM βCD, 1 mM AgNO_3_ and 7.5 mM NaOH. The Ag^+^/βCD molar ratio of this colloid remained 1:2.25, but it was ten times more diluted. 

All the synthesized colloids were stable for several days on the bench and several weeks if stored under refrigeration.

### 2.3. SERS Measurements

A standard 1 cm-path-length quartz cuvette was used in the SERS measurements. Samples for SERS experiments were prepared by adding 20 µL of 0.5 M Na_2_SO_4_ aqueous solution to 1000 µL of the AgNPs. This electrolyte solution is often employed to activate the surface of AgNPs, inducing their aggregation, as already shown in previous work [34]. Then, 10 µL of the sample was added. SERS spectra at different concentrations of the sample were recorded in order to find the minimum concentration that could be detected by this technique.

### 2.4. Instrumentation

Zeta potential and dynamic light scattering (DLS) measurements were performed in a Zetasizer Nano ZS (Malvern Panalytical Inc. Westborough, MA, USA) at a 632.8 nm laser line, the sample being placed in a polystyrene disposable cuvette. Transmission electron microscopy (TEM) images of all the synthetized AgNPs were recorded on a microscope model Talos F200X (Thermo Fischer Scientific, Waltham, MA, USA), working at 200 kV. The images were acquired with a CMOS 16 Mpx camera Ceta 16M. The ImageJ 1.x software [35] was employed in the analysis of the number and size distribution of the NPs. UV-visible absorption spectra were recorded with a Cary 7000 spectrophotometer (Agilent, Santa Clara, CA, USA), using a 10 mm-pathlength quartz cuvette.

The SERS spectra were recorded by using an Invia Qontor Raman Confocal System (Renishaw, Wotton-under-Edge, Gloucestershire, UK) coupled to a Leica Microscope with the spectral resolution set at ±2 cm^−1^. An objective of 50× magnification in a macro configuration with a focal distance of 30 mm (NA 0.17) and 2400 L/mm holographic grating were used at 532 nm excitation.

## 3. Results and Discussion

### 3.1. Characterization of the AgNPs

Figure 1 shows the UV-visible absorption spectra of all the synthesized AgNPs. Most of them show absorption maxima at about 419 nm, except for Ag@Citr and Ag@BH NPs, where they are blue-shifted up to 408 and 389 nm, respectively. In addition, all the Ag@βCD colloids show a narrower full width at half maximum (FWHM) than the other ones, the Ag@βCD3 NPs showing the narrowest bands among the Ag@βCD colloids, which is due to the narrow dispersity of the NP sizes as shown below.

The effect of reducing the concentrations of the reagents on the shape, size and dispersion of the AgNPs was analyzed through their TEM images and respective histograms (Figure 2). All the NPs had a spherical shape, regardless of the reducing agent, but showed different diameters and dispersity, as can be seen in the histograms based on 70–200 NPs. The Ag@Citr and Ag@BH sols had smaller average diameters, about 18 and 12 nm, respectively, than Ag@HX and Ag@βCD3, with about 42 and 38 nm, respectively. However, aggregates with higher diameters (over 200 nm) can be observed for the Ag@Citr and Ag@HX colloids.

The 84% and 68% of the Ag@BH (Figure 2a) and Ag@HX (Figure 2b) NPs showed diameters between 5 and 20 nm and 5 and 40 nm, respectively, while a low percentage of large particles (140–240 nm in diameter) was detected in the case of Ag@HX. The Ag@Citr NPs (Figure 2c) show a large dispersity, with a diameter range of 5–60 nm (40%), and the diameters of a low percentage of NPs (1.2%) were over 200 nm. Of the Ag@βCD3 NPs (Figure 2d), 68% had diameters between 20 and 45 nm.

The distributions of the non-functionalized AgNPs were broader than that of the Ag@βCD3 sol, which showed a Gaussian distribution with a maximum at 40 nm. The TEM images and histograms of the Ag@βCD1 and Ag@βCD2 colloids, as well as more images of the Ag@βCD3 NPs, are shown in the Appendix A. Although all the synthesized Ag@βCD NPs had similar sizes and shapes, the Ag@βCD1 and Ag@βCD2 NPs showed greater dispersity than the Ag@βCD3 colloid.

Table 1 shows the respective zeta potentials and hydrodynamic radii. Two hydrodynamic radii were measured for the non-functionalized Ag@Cit, Ag@BH and Ag@HX NPs due to the aggregation. Similar values were obtained for the Ag@Cit and Ag@HX colloids. One was larger, ca. 85 nm, and the other was smaller, about 12 nm. A larger value of 165 nm was obtained for the Ag@BH NPs, given that these NPs seemed to be surrounded by a dark halo, probably due to immobilized electrolyte. A hydrodynamic radius of about 65–85 nm was measured for all the Ag@βCD NPs since they were covered by an organic layer, which may have been surrounded by water clusters, avoiding the aggregation and, therefore, producing a small size. The smallest standard deviation was obtained for the Ag@βCD3 NPs.

Regarding the zeta potential, all the AgNPs showed a negative value of ca. −40 mV, which means they could be considered as NPs with moderate stability. The standard AgNPs, Ag@Citr and Ag@HX, showed slightly more negative values than the Ag@βCD ones. In this latter case, different precursor concentrations induced low changes in the surface charge. Above all, the Ag@βCD2 had a higher zeta potential (−34.4 mV) than the Ag@βCD1 and Ag@βCD3 sols (−38.6 and −39.4 mV, respectively). Ag@βCD3 NPs, with a Ag^+^/βCD molar ratio ten times more diluted than Ag@βCD2, showed a slightly more negative zeta potential, −39.4 mV, among all the Ag@βCD.

### 3.2. Comparative SERS Enhancement of AgNPs

Trans-cinnamic acid (TCA) was selected to investigate the effectiveness of colloids in enhancing SERS spectra. This molecule was chosen because it provides a strong SERS spectrum characterized by two strong bands recorded at about 1600 cm^−1^, and its hydroxyl derivate (trans-3-hydroxicinnamic acid) was previously studied by our research group [28].

Figure 3 shows the Raman spectrum of solid TCA and the SERS recorded on non-functionalized NPs at different concentrations of the adsorbate. The Raman and SERS spectra of TCA are dominated by two strong bands recorded at about 1635 and 1600 cm^−1^ and assigned to the C=C stretching and 8a ring-stretching modes, respectively. Two medium-intensity SERS bands were also recorded at about 1250 and 1000 cm^−1^, assigned to ring-substituent C_benzene_–C_ethylene_ bond stretching and 12 ring-deformation normal modes, respectively. Small wavenumber shifts are observed due to the adsorption effect. 

Appendix A shows the experimental vibrational wavenumbers and the corresponding assignment of the main bands of TCA according to previous studies as well as DFT (Density Functional Theory) calculations at the B3LYP/6-31G* level [28,36]. It is interesting to note that a weak broad band, recorded at about 1380 cm^−1^ and assigned to carboxylate symmetric stretching, appears in all the SERS spectra, which is not observed in the Raman spectrum. This result indicates that the adsorbed chemical species correspond to the cinnamate anion [36] linked to the metal through the carboxylate group, as already found in the SERS of other organic acids [28,34].

The lowest TCA concentrations detected by SERS were 5, 500 and 50 µM on the Ag@BH, Ag@HX and Ag@Citr sols, respectively. Although the citrate-reduced colloid was ineffective for the detection anions due to the residual citrate ions adsorbed to the AgNPs [37], it provided a better signal than the hydroxylamine-reduced colloid, which usually shows high sensitivity [32]. These limits of detection are of the same order of magnitude of those with other analytical techniques [38] employed in the detection of polycyclic aromatic compounds (PAHs), but are higher than those measured in SERS when using other functionalized AgNPs with viologen cations [16,17] and calixarenes [21], for instance, also used in detecting PAHs, or with βCD, as can be seen below for TCA.

Figure 4a–d show the SERS spectra of TCA on the Ag@βCD2 and Ag@βCD3 colloids, where the subtraction of the spectrum of the nanoparticles has been performed to reduce the intensity of the broad band of the colloid at 1410 cm^−1^. The spectra of the SERS on Ag@βCD1 are not shown because they have very low signal/noise ratios. This result could be related to a larger Ag^+^/βCD molar ratio than in the Ag@βCD2 and Ag@βCD3 sols. Therefore, the Ag@βCD1 colloid was disregarded as a SERS sensor for the here-studied molecules.

New enhanced bands were recorded in the SERS spectra of TCA on the Ag@βCD2 and Ag@βCD3 sols (Figure 4a,c) with significant intensity at 1378 and 1163 cm^−1^, and a shoulder at 1586 cm^−1^ is observed in the spectrum on Ag@βCD2 (Figure 4a). The band at 1379 cm^−1^ was recorded in both spectra, with a stronger intensity than in the case of non-functionalized NPs at 1388 cm^−1^. These differences suggest that TCA adsorbs through the carboxylate group. 

The appearance of new bands and the changes in the relative intensities and wavenumber shifts observed in the SERS on the Ag@βCD substrates could be due to the inclusion of TCA in the host βCD, producing a differentiated linking of the carboxylate group to silver surface atoms (non-modified sols) or the formation of a hydrogen bond with hydroxyl groups of βCD (modified sols), as was shown in previous work where the molecule was encapsulated within the CD cavity [39]. This inclusion would allow obtaining TCA close to the surface and favor a selective enhancement of particular bands, as happens for other aromatic carboxylic acids [34].

The lowest concentration of TCA detected by using the modified Ag@βCD2 and Ag@βCD3 NPs was 0.5 µM in both cases, almost ten times lower than the detection limit (5 µM) shown by the best standard Ag@BH colloid. The aging effect of the cyclodextrin sols on SERS signal was evaluated after five days as shown in Figure 4b,d with the aim of checking the lifetimes of these new SERS substrates. A differentiated behavior can be seen. Aging drastically favors the enhancement of the spectra recorded on Ag@βCD3, while a very poor spectrum was obtained in the case of the Ag@βCD2 sol. It can be seen that it is necessary to use a higher concentration, 1 µM, to detect the typical bands of TCA when using Ag@βCD2. Therefore, the SERS sensitivity is dependent on the aging of the colloids having a negative effect on Ag@βCD2 and an enhancement effect on Ag@βCD3.

Summarizing, although both the Ag@βCD2 and Ag@βCD3 sols can detect lower concentrations of TCA than non-functionalized colloids, only Ag@βCD3, with reactants ten times more diluted than in Ag@βCD2, is able to detect TCA concentrations in the order of ppb. This colloid was then selected among the studied Ag@βCD sols for detecting the IBU and CAF pollutants.

### 3.3. SERS Detection of ECs: Ibuprofen (IBU) and Caffeine (CAF)

The SERS spectra of the IBU and CAF pollutants were recorded at different concentrations, from 5.0 mM down to 0.1 mM, in order to find out the minimum concentration that could be detected by using Ag@βCD3 as well as non-functionalized NPs. The SERS spectra of concentrations higher than 5.0 mM were also checked, but they only showed stronger signals with similar relative intensities to those recorded at 5.0 mM. Figure 5 and Figure 6 show the same information: the SERS spectra of each contaminant employing the four types of AgNPs at the concentrations labeled in the inset and the Raman spectra of the respective 0.1 M aqueous solutions, which are placed at the bottom. Only the subtraction of the colloidal SERS signal from the SERS spectra obtained with the Ag@Citr and Ag@CD3 NPs was performed. The original SERS spectra of IBU and CAF together with the SERS of the pure colloids are shown in Appendix A. The experimental wavenumbers and vibrational assignments of both molecules are shown in Appendix A. B3LYP/6-31G* force field DFT calculations were carried out for supporting the empirical assignment based on previous studies for these molecules [40,41,42] and related ones [43,44].

The SERS spectra of CAF (Figure 5) show a striking feature. Those recorded on the Ag@HX NPs (Figure 5b) show a similar spectrum to that of the aqueous solution, characterized by a strong band recorded at 1324 cm^−1^ assigned to imidazole trigonal ring stretching, and three medium bands at 1714, 1605 and 555 cm^−1^ assigned to the C=O stretching+8a mode of pyrimidine, the stretching of the CC bond between the two rings, and pyrimidinic ring breathing, respectively [12,40]. However, the other SERS spectra (Figure 5a,c,d) showed differentiated spectra in which new bands, at 1680, 1250, 1008, 695, 647 and 507 cm^−1^, were enhanced with respect to the Ag@HX results. The first band is assigned to C=O stretching, with a high contribution of the CN bond stretching of pyrimidine, similar to the 8a mode in benzene-like molecules. The remaining ones are assigned to in-plane normal modes, with a motion mainly focused in the pyrimidinic part of the molecule, except those recorded at 1251 and 695 cm^−1^ that are assigned to in-plane and out-of-plane CH bonds of the imidazole ring [12,43,44] (see Appendix A).

This dual behavior of the SERS of CAF was already reported by other researchers when the spectra are recorded at different pH values [40,41]. Different explanations have been proposed on the basis of the adsorption centers, the adsorbed chemical species and/or the molecular reorientation induced by the pH, or the participation of different enhancement mechanisms of SERS.

Unlike other xanthines, CAF does not present tautomeric forms in aqueous solution because the nitrogen atoms are methylated. Therefore, only two chemical species may exist corresponding to the neutral molecule CAF and its protonated form in the free imidazole nitrogen CAF-H^+^. CAF has a pKa of 10.4 [41], and, therefore, the majority species in a neutral medium is the neutral form of CAF, but both neutral and protonated forms might have existed in our experiments, given that the pH of all the colloids was about 10. The SERS of CAF on Ag@HX was similar to that reported in acidic media, while the spectrum at a basic pH matched with that recorded on Ag@BH, Ag@Citr and Ag@βCD3 [41].

Alternatively, the two types of SERS could be consequences of the participation of physical and chemical enhancement mechanisms in the SERS signal. The physical plasmonic mechanism (also called electromagnetic) is the main mechanism responsible for the overall enhancement of SERS spectra, just as it happens in the case of Ag@HX NPs (Figure 5b), while the chemical mechanism could be responsible for the selective enhancement of particular bands [31,45]. The chemical contribution can involve changes in the relative intensities due to the subtle effect of the adsorption on the electronic structure of the molecule or to the presence of resonant processes up to new excited metal-to-molecule charge transfer (CT) states of the surface complex. In the CT mechanism of SERS, an electron is transferred from the metal to the molecule in the transient state, as has been demonstrated in the SERS of pyrazine [46,47], pyridine [48] and derivates [49,50]. The most enhanced bands of CAF at ca. 1680 and 1250 cm^−1^, which become the strongest ones in some spectr,a are assigned to the normal modes of pyrimidine and imidazole, υ(C=O) + 8a and CH stretching vibrations, respectively, resembling the normal modes that are enhanced in the SERS of pyridine or pyrazine through a charge transfer enhancement mechanism [46,47,48,49,50].

By considering the chemical interaction between CAF and the metallic surface, it can be concluded that this molecule adsorbs preferentially in an end-on orientation, as could be deduced on the basis of the propensity rules of the electromagnetic enhancement mechanism [51,52], given that no bands assigned to out-of-plane modes were enhanced. No evidence of the inclusion or interaction between CAF and βCD was visible, given that the SERS on Ag@BH, Ag@Citr and Ag@βCD3 was very similar. In any case, CAF could be detected at 0.5 mM with any non-functionalized NP, a concentration 10 times smaller than that previously reported [40,53], reaching an even lower limit (0.1 mM) when using the proposed Ag@βCD3 sol.

Unlike the SERS of CAF, a single behavior was observed in the SERS of IBU, irrespective of the AgNPs used (Figure 6). The Raman spectrum of IBU was mainly dominated by the band assigned to the 8a normal mode and recorded at about 1612 cm^−1^. This vibration was also strong in all the SERS, but these spectra are characterized by two strong bands recorded in the 1300–1400 cm^−1^ region (1390 and 1360 cm^−1^). Another, weaker SERS band was recorded at 887 cm^−1^, as can be seen clearly in Figure 6b, at a concentration of 5.0 mM. These three bands are assigned to mixed modes that contain contributions of carboxylate stretching and the deformation of methyl and methylene groups, υ_s_(COO) + δ_s_(CH_3_), δ(CH_2_) + δ_s_(CH_3_) and r(CH_2_) + r(CH_3_), respectively. The enhancement of the carboxylate stretching band (1390 cm^−1^) in a SERS measurement points to a molecular adsorption through this functional group, as it occurs in related molecules [34] such as TCA. Moreover, the 1358 and 887 cm^−1^ bands involve the symmetric deformation of the methyl nearest to the carboxylate group, and, therefore, it should be located close to the metal surface, which could explain the enhancement.

The lowest concentration of IBU that can be detected in SERS could be correlated with the size of the AgNPs. Considering the above-mentioned characteristic bands of IBU, the SERS recorded on the Ag@HX NPs (Figure 6b), with a greater diameter than the other NPs, showed the least sensitivity. IBU bands could be only detected in the spectrum recorded at 1.0 mM. Better results were obtained using the Ag@BH and Ag@Citr sols (Figure 6a,c), making it possible to detect the adsorbate at concentrations of 0.5 and 0.1 mM, respectively. Although the original SERS spectra on the Ag@βCD3 NPs (Appendix A) were not well resolved in the 1300–1400 cm^−1^ region, as happened with Ag@Citr (Appendix A), it was possible to observe the IBU bands at 0.1 mM after subtracting the colloidal signal (Figure 6c,d), providing the best limit of detection among all the discussed AgNPs. This value is of the same order of magnitude as the previously reported ones [39,42,54].

As in the previous case, no evidence of IBU–βCD host–guest interactions can be seen in the SERS, given that the intensities and wavenumbers of IBU seem to be insensitive to the colloid used.

## 4. Conclusions

A new synthetic route for AgNPs sols is proposed using βCD as a reducing agent with a Ag^+^/βCD molar ratio significantly lower than that previously published [26]. These new Ag@βCD3 colloids show an improved SERS efficiency with respect to the original proposal as well as to other standard silver sols very often used in SERS such as citrate Ag@Citr, borohydride Ag@BH or hydroxylamine Ag@HX colloids. It has been demonstrated that the size of the Ag@βCD NPs synthesized by using diluted reactants shows a lower dispersion and limit of detection for trans-cinnamic acid (TCA), which was selected as a test molecule, reaching trace level detection at about 20 ppb. The SERS enhancement by this new Ag@βCD3 sol was checked by means of a comparative study of the SERS of two pollutants, IBU and CAF, recorded by using this set of four silver sols. The study points out that the best SERS substrates for detecting the IBU and CAF pollutants correspond to the new synthesized Ag@βCD3 NPs, reaching a limit of detection of 0.1 mM in both cases. Other information that one can extract from the analysis of the SERS spectra is related to the molecular adsorption and even to the enhancement mechanism that contributes to the SERS signal. According to the relative enhancement of the SERS bands, IBU and CAF could be adsorbed through the carboxylate group and the imidazolic nitrogen and/or the carbonyl group, respectively, giving a perpendicular orientation with respect to the metallic surface. In addition, the charge transfer enhancement mechanism of SERS could be involved in the spectra of CAF recorded in colloids, where the bands assigned to mode 8a were enhanced. No evidence of the molecular inclusion of IBU and CAF in the cavity of βCD was detected.

In conclusion, a simple method for synthesizing AgNPs is proposed with an improved SERS detection of the CAF and IBU contaminants as compared to other standard colloids usually used in SERS.

## Figures and Tables

**Figure 1 nanomaterials-10-02339-f001:**
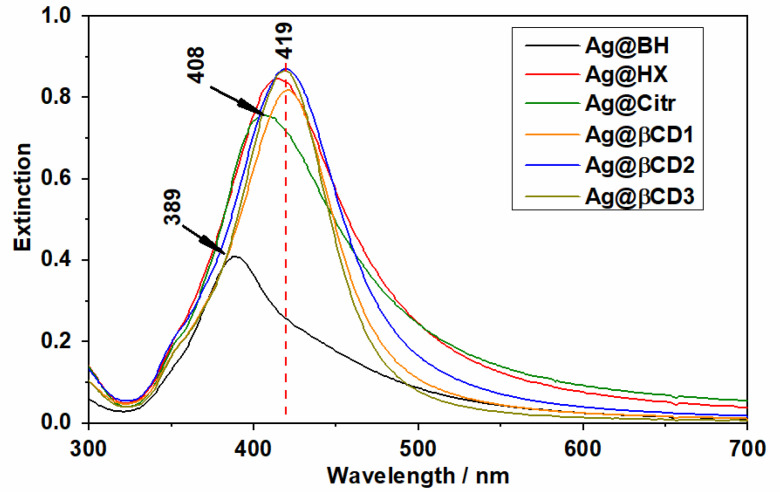
UV-visible absorption spectra of the different synthesized Ag nanoparticles (AgNPs).

**Figure 2 nanomaterials-10-02339-f002:**
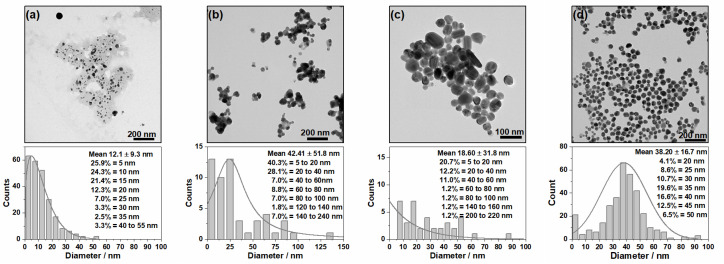
TEM images (above) and histogram distribution (below) for the different synthesized NPs: (**a**) Ag@BH, (**b**) Ag@HX, (**c**) Ag@Citr and (**d**) Ag@βCD3.

**Figure 3 nanomaterials-10-02339-f003:**
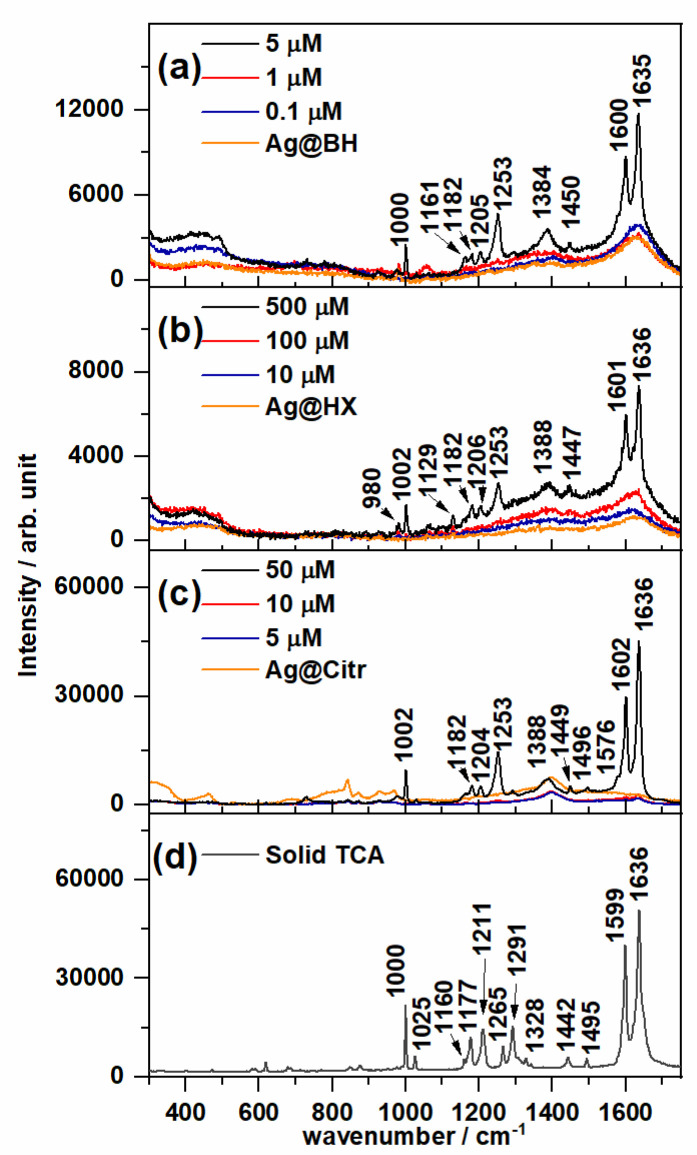
Surface-enhanced Raman spectroscopy (SERS) spectra of trans-cinnamic acid (TCA) recorded at different concentrations using (**a**) Ag@BH, (**b**) Ag@HX and (**c**) Ag@Citr NPs. The respective Raman spectra of colloids are also shown. (**d**) Raman spectra of solid TCA.

**Figure 4 nanomaterials-10-02339-f004:**
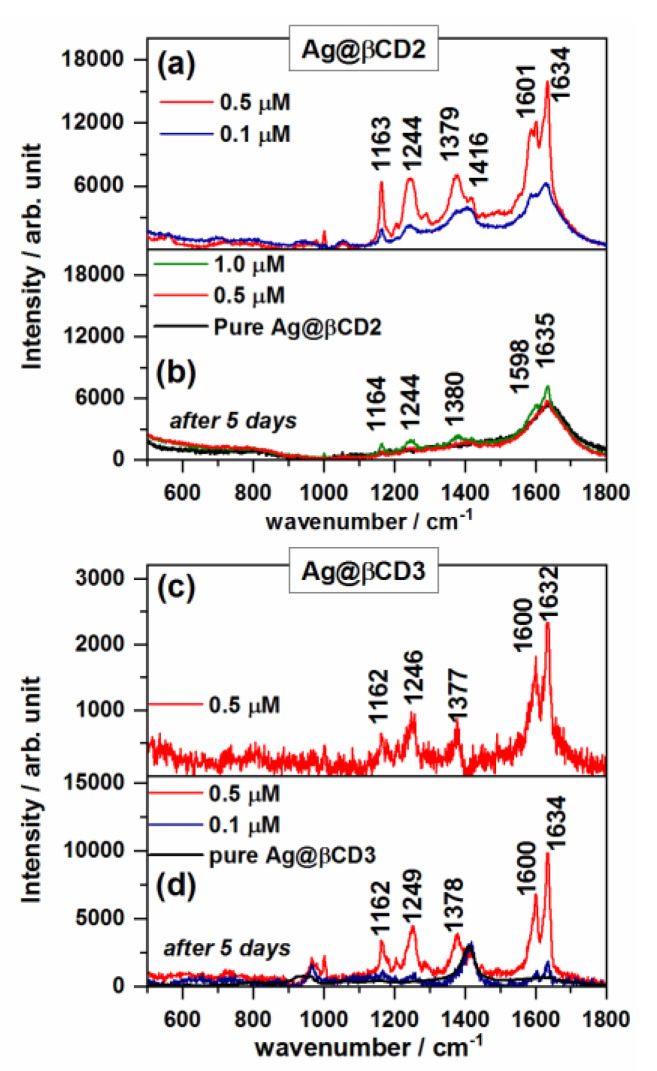
SERS spectra of TCA recorded at different concentrations on (**a**) Ag@βCD2 and (**c**) Ag@βCD3 colloids. The aging effect on the SERS signal after five days is shown in (**b**), (**d**), respectively.

**Figure 5 nanomaterials-10-02339-f005:**
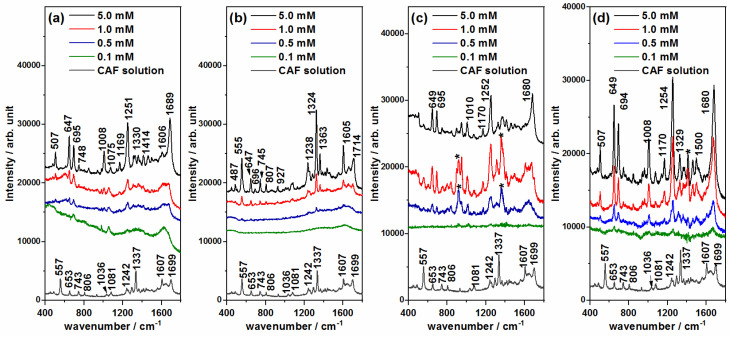
SERS spectra of caffeine (CAF) recorded at different concentrations using (**a**) Ag@BH, (**b**) Ag@HX, (**c**) Ag@Citr and (**d**) Ag@βCD3 NPs. Raman spectrum of 0.1 M aqueous solution at the bottom. Colloidal bands after subtraction are marked with (*).

**Figure 6 nanomaterials-10-02339-f006:**
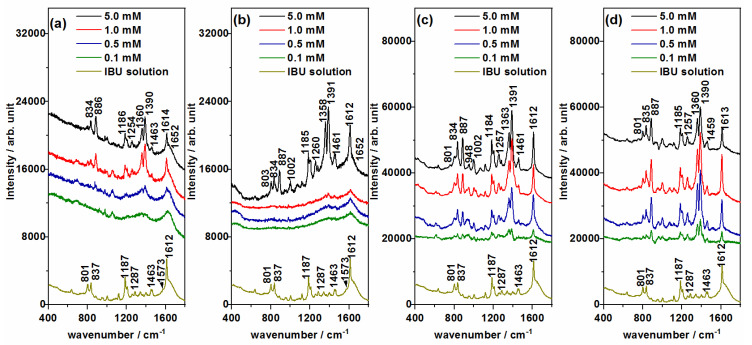
SERS spectra of ibuprofen (IBU) recorded at different concentrations using (**a**) Ag@BH, (**b**) Ag@HX, (**c**) Ag@Citr and (**d**) Ag@βCD3 NPs. Raman spectrum of 0.1 M aqueous solution at the bottom.

**Table 1 nanomaterials-10-02339-t001:** Zeta potential and hydrodynamic radius for the synthesized AgNPs.

Colloid	DLS (Hydrodynamic Radius)		Zeta Potential/mV
	Peak 1/nm	Peak 2/nm	
Ag@BH	166 ± N.A. *	3.53 ± 1.41	−37.3 ± 9.70
Ag@HX	84.3 ± 42.1	11.8 ± 3.76	−41.3 ± 11.0
Ag@Citr	83.4 ± 41.0	11.4 ± 3.59	−42.7 ± 12.0
Ag@βCD1	71.6 ± 31.9	-	−38.6 ± 13.1
Ag@βCD2	85.6 ± 50.1	-	−34.4 ± 13.7
Ag@βCD3	66.2 ± 26.9	-	−39.4 ± 11.4

* Standard Deviation not applicable.

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
