# Peer review of "Comparative Performance of Citrate, Borohydride, Hydroxylamine and β-Cyclodextrin Silver Sols for Detecting Ibuprofen and Caffeine Pollutants by Means of Surface-Enhanced Raman Spectroscopy"

_nanomaterials, 2020, doi:10.3390/nano10122339_

Round 1

Reviewer 1 Report

Comment to the Authors

 In this research, authors prepared various silver nanoparticles with citric acid, borohydride, hydroxylamine and b-cyclodextrin as reducing and surface coating materials and investigated their potential for pollutant detection by SERS. Data are clear. Thus, there is a worth to report, although the novelty and scientific interest are poor.

Major comments

  1. Page 5; Authors discussed about Zeta-potential of AgNPs. For example, Zeta-potential of Ag@bCD2 is higher than that of Ag@bCD1 and Ag@bCD3. I don't feel these are significant. If authors would like to discuss on this difference, they should show their experimental error or deviation.
  2. Page 7; Authors mentioned about the unique effects of aging. But there is not clear explanation on this mechanism.

Minor comments

  1. Page 2; There is a misspelling of "host-gest interaction".
  2. Please unify the style of writing on concentration. For example "1.5x10-2 M", "0.010 M "and "0.5 mM".
  3. Page 5; Authors showed hydrodynamic radius in Table 1. They need to think about a significant figure.

Reviewer 2 Report

he authors present a comprehensive report on a modified preparation of silver colloids produced by using cyclodextrin as a reducing agent and compare these to materials prepared by standard routes. They convincingly show that their materials offer lower limits of detection for their test molecules.
While the performance of the colloids is very good for pure analytes, what happens in a mixture? This is relevant because the authors are proposing the use of SERS as a "no pre-treatment process (line 55)". A simple test would be to measure the spectra from a 1:1 molar ratio of caffeine and ibruprofen.
Are their detection limits still valid for a mixture? If not, then it would suggest that for a real-world sample (which is likely to be more complex) the method will be of limited use.
I recommend the paper for publication in Nanomaterials subject to minor revision.

Minor points
Line 125: Surely "1:173" should be "1:1.73"?
There are numerous spelling mistakes (eg, line 65 "guess-host" line 312: "mayority", line 384 "evidences") and instances of awkward English (eg line 48: "as well as they are time-consuming., line 51 "is becoming an useful", line 157: "were used under 532 nm excitation."). The use of an English language checking service is recommended.
